# Another dengue fever outbreak in Eastern Ethiopia—An emerging public health threat

**Mulugeta Asefa Gutu**[1]*, **Alemayehu Bekele**[2], **Yimer Seid**[3], **Yusuf Mohammed**[4], **Fekadu Gemechu**[5], **Abyot Bekele Woyessa**[5], **Adamu Tayachew**[5], **Yohanis Dugasa**[5], **Lehageru Gizachew**[1], **Moti Idosa**[1], **Ryan E. Tokarz**[6], **David Sugerman**[6]

**1** Ethiopian Field Epidemiology Training Program, Addis Ababa, Ethiopia, **2** Ethiopian Public Health Association, Addis Ababa, Ethiopia, **3** Addis Ababa University, Addis Ababa, Ethiopia, **4** World Health Organization Country Office for Ethiopia, Addis Ababa, Ethiopia, **5** Ethiopian Public Health Institute, Addis Ababa, Ethiopia, **6** Centers for Disease Control and Prevention, Atlanta, Georgia, United States of America

* muluugetaaa@gmail.com

## Abstract

### Background

Dengue Fever (DF) is a viral disease primarily transmitted by *Aedes (Ae.) aegypti* mosquitoes. Outbreaks in Eastern Ethiopia were reported during 2014–2016. In May 2017, we investigated the first suspected DF outbreak from Kabridahar Town, Somali region (Eastern Ethiopia) to describe its magnitude, assess risk factors, and implement control measures.

### Methods

Suspected DF cases were defined as acute febrile illness plus $\geq$2 symptoms (headache, fever, retro-orbital pain, myalgia, arthralgia, rash, or hemorrhage) in Kabridahar District residents. All reported cases were identified through medical record review and active searches. Severe dengue was defined as DF with severe organ impairment, severe hemorrhage, or severe plasma leakage. We conducted a neighborhood-matched case-control study using a subset of suspected cases and conveniently-selected asymptomatic community controls and interviewed participants to collect demographic and risk factor data. We tested sera by RT-PCR to detect dengue virus (DENV) and identify serotypes. Entomologists conducted mosquito surveys at community households to identify species and estimate larval density using the house index (HI), container index (CI) and Breteau index (BI), with BI$\geq$20 indicating high density.

### Results

We identified 101 total cases from May 12–31, 2017, including five with severe dengue (one death). The attack rate (AR) was 17/10,000. Of 21 tested samples, 15 (72%) were DENV serotype 2 (DENV 2). In the case-control study with 50 cases and 100 controls, a lack of formal education (AOR [Adjusted Odds Ratio] = 4.2, 95% CI [Confidence Interval] 1.6–11.2) and open water containers near the home (AOR = 3.0, 95% CI 1.2–7.5) were risk factors, while long-lasting insecticide treated-net (LLITN) usage (AOR = 0.21, 95% CI 0.05–0.79)

**Data Availability Statement:** All relevant data are within the manuscript and its Supporting Information files.

**Funding:** The author(s) received no specific funding for this work.

**Competing interests:** The authors have declared that no competing interests exist.

was protective. HI and BI were 66/136 (49%) and 147 per 100 homes (147%) respectively, with 151/167 (90%) adult mosquitoes identified as *Ae. aegypti*.

## Conclusion

The epidemiologic, entomologic, and laboratory investigation confirmed a DF outbreak. Mosquito indices were far above safe thresholds, indicating inadequate vector control. We recommended improved vector surveillance and control programs, including best practices in preserving water and disposal of open containers to reduce *Aedes* mosquito density.

## Author summary

In 2017 an outbreak of Dengue fever (DF) was reported in Kabridahar Town, Ethiopia. This mosquito transmitted disease was recently detected in Ethiopia only four years prior, with this being the first time it was identified in the area. In response, our team was dispatched to confirm the presence of the disease, identify potential causes, and implement mitigation and control measures. We identified and compared suspected cases and suspected non-cases to identify the potential risk factors of infection. Laboratory confirmation of infection and disease-type was also performed. Due to the entomological nature of disease transmission, additional entomological investigations were conducted at the households of both groups to understand its influence at the household level. Through these measures, we were able to establish the presence of DF in Kabridahar Town and identify risk factors leading to infection. Risk factors included a lack of formal education and open water containers near the home, while the presence of long-lasting insecticide-treated nets were found to be protective. Mitigation and control measures were implemented to combat or promote the identified risk and protective factors respectively. Cases counts began to reduce five days after the onset of these measures. Recommendations were made based on our findings to prevent future outbreaks. The last case was recorded ten days after implementation of the mitigation and control measures.

## Introduction

Dengue fever (DF) is a viral disease primarily transmitted by *Aedes (Ae.) aegypti* mosquitoes and is found in tropical and sub-tropical climates worldwide, mostly in urban and semi-urban areas [1,2]. The disease is commonly characterized by a rapid onset of fever, headache, rash, and severe joint and muscle pain [3]. Repeated infections with different serotypes can cause a life-threatening condition called severe dengue [4]. Globally, DF incidence has increased 30-fold over the last 50 years, with increasing geographic expansion to new countries and, in the last decade, from urban to rural settings [5]. DF is endemic throughout the tropics and sub-tropics, with an estimated 3.8 (95% confidence interval [CI] 3.5–4.1) billion people (roughly 53% of the global population) living in areas that are suitable for Dengue virus (DENV) transmission with the vast majority in Asia, followed by Africa and the Americas [6], with models estimating upwards of 400 million infections annually [4].

Known risk factors for dengue virus infection include those that facilitate mosquito breeding sites, such as extended rainfall and high humidity and high temperature [7,8]. Having indoor and outdoor water containers—especially uncovered containers such as buckets, drums, tires, pots, and jerry cans—with stagnant water provides breeding sites to for *Ae.*

*aegypti* in urban areas and facilitates transmission [9,10]. The absence of door and window screens and failure to use repellents has also been shown to facilitate infection [11,12]. This is particularly important, as the weather conditions in Ethiopia's Somali region results in the commonly observed practice of early morning and daytime sleeping.

Despite being considered a neglected tropical disease, DF is a substantial issue on the African continent. During 1960–2010, 22 countries in Africa reported sporadic DF cases or outbreaks, with most resulting from serotype 2 (DENV 2) [13,14]. A recently published systematic review looking at 80,977 participants from 76 studies during the years of 2000–2017 provides strong supports for this idea. This meta-analysis of the prevalence of DENV infection in African residents illustrates high prevalence on the continent, although with heterogeneity depending on location, disease presentation, and viral markers [15]. Immunoglobulin and RNA analysis for example, showed Western Africa to have the highest IgG seroprevalence in febrile participants, while Central Africa displayed the highest IgM seroprevalence and RNA prevalence in this same participant group. Geographic differences were also shown upon analysis of apparently healthy individuals [15].

In 2013 DF was first detected in Ethiopia, with over 12,000 DF cases reported [16]. Since this time, outbreaks have been confirmed in the Northern and Eastern parts of the country. The first confirmed outbreak occurred in Dire Dawa, demarcated as both a city and a region and located central-eastern Ethiopia. The outbreak impacted over 11,000 people [17]. The following year, in 2014, outbreaks occurred again in Dire Dawa, as well as in the south-eastern portion of the country in Godey Town, Somali Region and Adaar Woreda in Afar Region, which is located in northern Ethiopia [17,18,19]. Outbreaks have since occurred on a yearly basis in Godey Town [19] and Dire Dawa [20], indicating that the virus has become well established within these portions of the country. Additional evidence of DF in northern Ethiopia was detected in Humera, Tigray Region and Metema, Amhara Region via a serological cross-sectional study of febrile patients in these regions [21]. This same study found anti-DENV IgG seropositivity during each month of the year, suggesting the potential of DENV endemicity.

On May 13, 2017, the Somali regional Health Bureau notified the Ethiopian Public Health Institute (EPHI) / Public Health Emergency Management (PHEM) of the first suspected DF outbreak in Kabridahar Town (Korahey Zone, in Eastern Ethiopia), and requested an investigation. A team comprising a field epidemiologist, environmental health expert, medical doctor, and laboratory technologist deployed to confirm the outbreak was due to dengue virus, assess risk factors for dengue, perform an entomological survey, and provide recommendations for outbreak mitigation and control.

## Methods

### Ethics statement

The Research Ethical Review Committee of Somali Regional Health Bureau approved the study and provided ethical clearance. The national public health authorities are empowered and mandated to conduct outbreak investigations as indicated in the Ethiopian Public Health Institute Establishment Council of Ministers Regulation No.301/2013 (*Federal Negarit Gazette of the Federal Democratic Republic of Ethiopia. Ethiopian Public Health Institute Establishment Council of Ministers Regulation. Regulation no. 301/2013. 20th Year No. 10. Addis Ababa, Ethiopia; 2014*) to protect the community from outbreak adverse effects compelling the community to participate during outbreak investigations, as they are benefitted from the interventions.

Ethical clearance was also obtained from the Ethiopian Public Health Institute (EPHI). A letter was written from the Regional Health Bureau to obtain approval for data collection and

additional permission was sought from Kabridahar District Health Office. In all villages and towns, the investigators were accompanied by health extension workers who explained the purpose of the visit to the owner of the houses visited. Oral informed consent was obtained from the head of the households for larval and adult mosquito collection and from adult patients and caretakers of children. Assent was obtained from older children before participating in the study. Confidentiality was assured throughout the study. Protection of individual privacy was conducted using confidential codes and the analysis of the resulting outbreak data was conducted anonymously.

## Study area

Kabradihar District lies within the Korahey Zone of the Somali region. The District includes Kabradihar Town, comprising 10 urban villages and 16 rural villages surrounding the town. The town is nearly 400 km from Jijiga, the capital of the Somali region, bordered by rural lands, at an elevation of 493 meters above sea level with an average temperature of 30˚ C. The total population of Kabradihar Town was estimated at 60,000 in 2017. The town has one district hospital, three health posts, and five private clinics [22].

## Study design

There is no ongoing surveillance system for arboviruses in Ethiopia. Cases were identified by reviewing medical records, line lists, and rumor logbooks, as well as door-to-door active searches with the assistance of community leaders and community health extension workers. Line lists refer to a recorded line list table that summarizes information about persons who may be associated with an outbreak, while rumor logs are records of information passed along from the community, notifying of an unusual event.

We defined suspected DF as acute febrile illness of 2–7 days duration, with two or more of the following: headache, fever, retro-orbital pain, myalgia, arthralgia, rash, and hemorrhagic manifestations from May 4—June 22, 2017 in a resident of Kabridahar District. This time period begins one week prior to the first reported case and concludes 3 weeks after the final case in the outbreak was reported. We defined severe dengue as DF with evidence of severe hemorrhage, severe organ involvement, or severe plasma leakage. All persons identified through active surveillance were provided medical care. All suspected cases were sent to Kabridahar Hospital, the nearest general hospital, while complicated and suspected severe cases were sent to Jijiga referral hospital. A confirmed case was defined as detection of dengue virus (DENV) in blood by real time reverse transcriptase-polymerase chain reaction (RT-PCR) [23].

We conducted descriptive analysis on all 101 suspected cases, followed by a 1:2 case-control study, matched by neighborhood. For every identified suspected case (one per home), two asymptomatic neighborhood controls were selected from different residences and interviewed on the same day as the case. The sample size for the case-control study was calculated using a confidence level of 95%, power of 80%, a 50% prevalence of using an LLITN, an odds ratio (OR) of 2.98 [12], and with a 1:2 ratio of cases to controls for a total sample size of 150 (50 cases and 100 controls). The last 50 suspected cases identified were included in the case-control study. Attack rate was conducted using all cases, both confirmed and suspected, in relationship to the total population. Data was collected through face-to-face interviews with participants at the hospital or in their homes, using structured questionnaires.

## Data processing and analysis

Data was double-entered into Epi Info software version 7.2. We calculated descriptive statistics and conducted conditional bivariate and multivariate logistic regression to compare exposures

between cases and controls. Conditional regression was employed to account for matching. All variables with $p < .25$ in bivariate analysis underwent further conditional multivariate logistic regression. Adjusted odds ratios (AOR) with 95% confidence intervals (CI) are presented. Possible a priori geographic confounders were accounted for through the neighborhood-matching component of the case-control study design.

## Laboratory investigation

Serum specimens of 3-5ml were collected into a serum separator tube from a subset of 21 cases and shipped with their accompanying case investigation forms to the National Reference Laboratory at Ethiopian Public Health Institute (EPHI). All samples were centrifuged at health facilities within the outbreak area before being shipped to the National Influenza and Arbovirus Laboratory at EPHI, as there were no established local and regional capacities for laboratory detection of dengue viruses. Appropriate implementation of triple packaging of the samples to maintain the cold chain was ensured and each completed case reporting form was also included.

The serum was then subjected to RNA extraction using QIAGEN RNA extraction Mini kit (QIAamp Viral RNA Mini Kit). Amplification was conducted using the Invitrogen SuperScript III Platinum One-Step qRT-PCR Kit. The total reaction volume was 25ul and composed of 10ul of the RNA extract elute and 15ul of the prepared master mix from the Invirtrogen kit (SuperScript III). Each sample was simultaneously tested for dengue (DENV), zika (ZIKV) and chikungunya (CHIKV) viruses using the Trioplex Real-time RT-PCR Assay [24]. The Trioplex Real-time RT-PCR Assay was developed for simultaneous identification of DENV, CHIKV, and ZIKV at a single RT-PCR well by assigning a different detection channel and human specimen control at a separate well for each sample (FAM for DENV, VIC for CHIKV, TEXAS RED for ZIKV and FAM for HSC). The corresponding thermocycler conditions was adjusted as follows: reverse transcription to cDNA at 45˚C for 10 min, Taq activation and denaturation at 95˚C for 10 min, and amplification at 45 cycles of 95˚C for 15 seconds, 55˚C for 60 seconds (extension and data collection). Specimens were considered positive if exponential curves with logarithmic, linear, and plateau phases were produced. Concurrently, a valid individual human specimen control (extraction control), along with positive and negative controls must have also been present. As per the kit manufacturer's recommendation, amplification curves with cycle threshold values $\leq 38$ from the total reaction cycle of 45 were considered to be positive.

All samples identified as DENV positive were then retested for differentiation of the four dengue serotypes. The dengue positive samples were selected and analyzed for serotype using the fast track diagnostics (FTD) kit strictly following the manufacturer's procedure [25]. For this process, a new mastermix preparation and thermocyler program was utilized. The mastermix contained mixes of primer and probes of Dengue serotype 1–4 within the same tube but designated to different detection channels. Detection channels FAM, JOE, Cy5, and ROX were designated for Dengue virus type 1, 2, 3, and 4 respectively. A total reaction volume of 25ul, 15ul mastermix and 10ul of the extracted RNA elute, was utilized. The corresponding thermocycler conditions was adjusted as follows: reverse transcription to cDNA at 50˚C for 15 min, Taq activation and denaturation at 94˚C for 1 min, and amplification at 45 cycles of 94˚C for 8 seconds, 60˚C for 60 seconds (extension and data collection).

## Entomological investigation

The entomological investigation occurred within all villages in Kabridahar Town from May 14–27. All households with suspected cases and the majority of those in the control group

from the case-control study were investigated. Investigations were conducted in a house-to-house fashion, observing for mosquito breeding sites at water sources and in homes, as well as for the presence of uncovered water containers, screening of windows, and empty containers inside and outside the homes. Villages were selected based on consultation with the Zonal PHEM coordinator and Kabridahar Town Health Office and primarily involved villages where suspected cases originated. Purposive sampling of potential breeding sites and indices occurred where cases were reported and were extended up to a 100-meter radius. Sampling continued up to three kilometers from town, ending with the furthermost residences. Demographic and entomological data on sampled houses were compiled using standard data collection forms.

All identified water-holding containers, both inside and outside of homes, were visually examined for immature stages of mosquitoes. Adult mosquitoes were collected from households where suspected cases of DF originated and bordering villages by aspiration and pyrethrum spray sheet collection. Larval sampling covered the domestic and peri-domestic environments to estimate risk indices. In rural areas, water samples collected in plant leaf axils were inspected for larva and pupae of *Aedes*. All collected larvae, pupae, and adult mosquitoes were placed in labelled paper cups and transported to a temporary entomology laboratory in Kabridahar Hospital. Specimens in the juvenile aquatic stage were reared to adulthood and then identified to the species level [26]. The investigation team also observed the sanitation practices, water collection habits, drainage systems, and personal protection measures against mosquitoes.

Trained entomologists utilized key morphological characteristics to identify collected mosquito specimens to the genus level. Those identified as *Aedes* mosquitoes were identified to be of two species, *Ae. aegypti* and *Ae. africanus*, using a standard morphological identification key developed by Rueda, 2004 [26]. All hatched adults reared in the temporary laboratory were identified to the species level using this same methodology.

Analysis of the standard *Aedes* larval indices, such as house index (HI), container index (CI), and Breteau index (BI), were carried out. The HI is widely used for measuring population levels, while CI provides the proportion of water-holding containers that demonstrate larval activity. The BI is considered most informative as it establishes the relationship between the house itself and its associated positive containers [27]. The surveillance guidelines of China [28] reported the threshold for the control of dengue transmission at a BI output below 5, with values of 5–10 indicating a risk of transmission. BI readings of 10–20 and >20 were shown to be indicative of an outbreak, and regional epidemic, respectively. During an outbreak the outputs range of importance may reduce, as a BI of $\geq 4$ has been shown to be predictive of transmission during such conditions [29].

The following formulas were used to determine these indices [30].

$$\text{HI} = \frac{\text{Number of houses with immature mosquitoes}}{\text{Number of households inspected}} \times 100$$

$$\text{CI} = \frac{\text{Number of water-holding containers with immature mosquitoes}}{\text{Number of water-holding containers}} \times 100$$

$$\text{BI} = \frac{\text{Number of water-holding containers with immature mosquitoes}}{\text{Number of households inspected}} \times 100$$

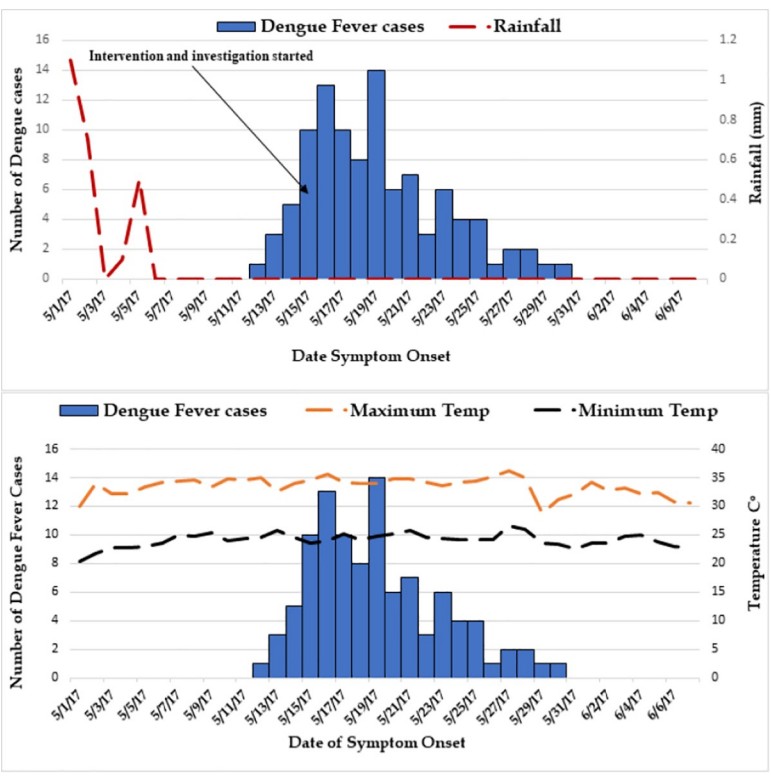

**Fig 1. Daily Dengue Fever Cases and Climatic Factors–Kabridahar District, Korahay Zone, Somali region, Ethiopia, May to June, 2017.**

## Results

A total of 101 DF suspected cases and 1 death were reported from the 10 villages of Kabridahar Town, with onset dates between May 12—May 31, 2017 (Fig 1). No cases were reported from the surrounding villages. The overall attack rate (AR) was 17/10,000 with a case-fatality rate (CFR) of 1%, with males making up the majority of cases (61%). The median age of DF suspected cases was 25 years (range: 4–65 years). Persons aged 15–45 were the most affected (AR = 30.3/10,000 population), followed by persons >45 years of age (AR = 28.2/10,000 population). The least-affected age group were those less than 5 years (AR = 1.4/10,000 population) followed by persons aged 5–14 years (AR = 16.7/10,000 population).

The highest attack rate, 35 per 10,000 population, was seen in Village 07 and was followed by Village 01, with 23 per 10,000 (Fig 2A). The first cases appeared on May 12th in Village 07,

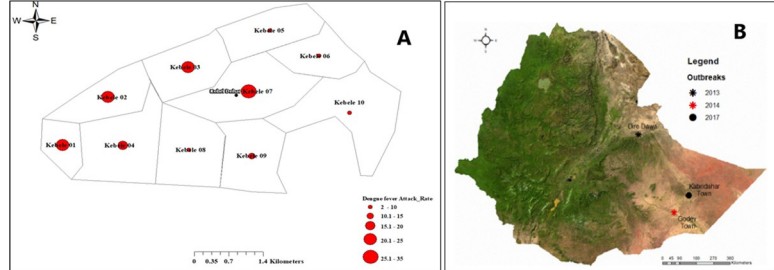

**Fig 2. Distribution of Dengue Fever.** (A) Attack Rate (Cases per 10,000 population) by Village–Kabridahar Town (Villages 1–10), Kabridahar District, Korahay Zone, Somali region, Ethiopia, 2017. (B) Important Dengue Fever outbreak locations since first detected in Ethiopia in 2013. (*Basemap obtained from https://www.worldofmaps.net/en/africa/map-ethiopia/satellite-map-ethiopia.htm and is compatible with CC BY licensing*).

the most population-dense of the villages. The index case, a male age 51, was on recorded May 12[th] in Village 07. The patient presented to a district hospital and recovered. DENV infection was later confirmed positive by RT-PCR. The patient had no travel history beyond Kabridahar Town and as a result, the investigation and response were initiated. Shortly after identification of the index case, infections appeared in all nine other villages in Kabridahar Town, with a concentration occurring in the four westernmost villages (01–04). The peak number of cases during the outbreak occurred on May 19[th] (Fig 1).

All suspected cases had fever; other symptoms included joint pain (96%), myalgia (94%), headache (88%), retro-orbital pain (74%), and skin rash (33%). Laboratory testing was not available at the outbreak location and shipping specimens for outside analysis was often not feasible. As a result of these resource limitations, only 21 serum samples were able to be tested. From the subset of 21 suspected cases, 15 (72%) were positive for dengue virus serotype 2 (DENV 2) by RT-PCR. None of the samples tested positive for DENV 1, DENV 3, DENV 4, Zika virus, or Chikungunya virus. Five of the cases presented severe dengue warning signs, demonstrating epistaxis or gingival bleeding, and one of which presented with thrombocytopenia. Of these five cases, three were referred to Jijiga Regional Referral Hospital. In hospital, one patient, age 49, was treated with intravenous fluid therapy, electrolyte replacement and frequently monitored in an intensive care unit. This patient recovered without further complication and was discharged home. The remaining two patients, age 39 and 41, deteriorated hemodynamically and blood was transfused. The 39 years-old patient recovered post-transfusion without further complication and was discharged home, while the 41 years-old patient died. An autopsy was not conducted on the deceased patient to identify cause of death, however this patient, as well as the other 4 warning signs-presenting cases tested positive for dengue.

In the case-control study, results from bivariate and multivariate analyses are displayed in Table 1. Variables from the bivariate analysis with p-value <0.25 underwent further conditional multiple logistic regression. Outputs from this regression showed that having no formal education (AOR = 4.23; CI 1.60–11.17), having open containers inside and outside home (AOR = 3.02; CI 1.22–7.48), having larvae identified in household containers inside or outside the house (4.17; CI 1.66–10.51), and reporting daily wearing of short-sleeve t-shirts (AOR = 3.29; CI 1.29–8.39) increased the odds of being a suspected dengue cases, while long-lasting insecticidal net (LLITN) use the previous night (AOR = 0.21; CI 0.05–0.79) was protective (Table 1).

From May 14–27, 2017, a total of 136 houses were included in the entomologic survey. The remaining 14 houses were temporary shelters set up by the often-nomadic pastoralist community in the area. The team inspected 411 water-holding containers (average of 3 per house) and detected *Ae. aegypti* (n = 151) and *Ae. africanus* (n = 16) in 210 (51%) containers (CI = 51%) and 66 of 136 households (HI = 49%). Among the 136 homes inspected, immature mosquitoes were found in 210 water holding containers (BI = 154). Further detail from the entomological survey is shown in a supporting informational table (S1 Table). A majority (60%) of mosquito breeding sites identified were large containers, such as cement tanks (i.e., birka), jerry cans, and buckets; but breeding sites were also identified in discarded items (i.e. tins, plastic bottles, and tires), which were common in the affected communities.

In response to the epidemic, from May 15—June 17, 2017, we conducted an active case search, distributed long lasting insecticide-treated nets (LLITNs), collected discarded water-holding items, and initiated public health communication about DF in all affected villages and border areas. We also provided DF case-management training to healthcare providers at all health posts and health centers. Five days after starting these interventions, case counts began to fall, with the last case 10 days later (Fig 1). Rainfall also significantly declined just prior to,

**Table 1. Bivariate Conditional and Multivariable Logistic Regression Analysis: Socio-demographic Characteristics and Potential Risk and Protective factors for Dengue Fever (DF)—Kabridahar District, Korahay Zone, Somali region, Ethiopia, 2017.**

| Risk/Protective Factor | Case (N = 50) | Control (N = 100) | COR (95%) CI) | AOR (95% CI)* |
|---|---|---|---|---|
| **Sex** | | | | |
| Female (ref) | 16 | 35 | 1.14 (0.55–2.35) | — |
| Male | 34 | 65 | | |
| **Age group** | | | | |
| 0–14 years (ref) | 4 | 9 | 1.22 (0.33–4.22) | — |
| 15–44 years | 40 | 74 | 0.79 (0.175–3.585) | |
| ≥45 years | 61 | 17 | | |
| **Marital status** | | | | |
| Single (ref) | 22 | 43 | 0.96 (0.48–1.90) | — |
| Married | 28 | 57 | | |
| **Ethnicity** | | | | |
| Somali (ref) | 15 | 27 | 0.86 (0.40–1.82) | — |
| Other | 35 | 73 | | |
| **Occupation** | | | | |
| Daily laborer (ref) | 19 | 25 | 1.20 (0.15–9.48) | — |
| Government employee | 2 | 2 | 0.87 (0.30–2.53) | |
| Farmer pastoralist | 10 | 15 | 1.42 (0.51–3.91) | |
| House wife | 15 | 14 | 0.58 (0.05–6.67) | |
| Merchant | 1 | 4 | 0 | |
| Other | 0 | 23 | 0.25 (0.06–1.0193) | |
| Student | 3 | 17 | | |
| **Status of containers inside and outside home** | | | | |
| Closed (ref) | 16 | 40 | 3.18 (1.55–6.52) | 3.02 (1.22–7.48) |
| Open | 34 | 60 | | |
| **Own a LLITN** | | | | |
| No (ref) | 11 | 9 | 0.36 (0.134–0.913) | — |
| Yes | 39 | 91 | | |
| **Use an LLITN** | | | | |
| No (ref) | 10 | 7 | 0.24 (0.08–0.69) | 0.21 (0.05–0.79) |
| Yes | 29 | 84 | | |
| **Formal education** | | | | |
| Yes (ref) | 13 | 54 | 3.34 (1.58–7.03) | 4.23 (1.60–11.17) |
| No | 37 | 46 | | |
| **Existence stagnant water around the home** | | | | |
| No (ref) | 45 | 94 | 1.74 (0.5–6.0) | — |
| Yes | 5 | 6 | | |
| **Presence of larvae in the container** | | | | |
| No (ref) | 18 | 64 | 3.16 (1.55–6.41) | 4.17 (1.66–10.51) |
| Yes | 32 | 36 | | |
| **Travel history (Two weeks)** | | | | |
| No (ref) | 3 | 5 | 0.82 (0.18–3.59) | — |
| Yes | 47 | 95 | | |
| **Household indoor residual spray (IRS) application (six months)** | | | | |
| No (ref) | 50 | 100 | 1.00 | — |
| Yes | 0 | 0 | | |
| **Clothing usually worn** | | | | |
| Trousers/body full dress (ref) | 17 | 58 | 2.68 (1.32–5.43) | 3.29 (1.29–8.39) |
| Short and T-shirt | 33 | 42 | | |
| **Mosquito repellent used on skin** | | | | |
| No (ref) | 42 | 73 | 0.51 (0.21–1.23) | — |
| Yes | 8 | 27 | | |
| **Mosquito repellent used in the house** | | | | |
| No (ref) | 28 | 55 | 0.96 (0.48–1.9) | — |
| Yes | 22 | 45 | | |

(*Continued*)

**Table 1.** (Continued)

| Risk/Protective Factor | Case (N = 50) | Control (N = 100) | COR (95%) CI) | AOR (95% CI)* |
|---|---|---|---|---|
| **Sleeping inside screened windows or doors** | | | | |
| No (ref) | 27 | 47 | 0.75 (0.38–1.49) | — |
| Yes | 23 | 53 | | |

**Closed container:** Tightly covered container, **COR**: crude odds ratio, **AOR**: adjusted odds ratio

*AOR calculated for significant CORs ($p<0.25$)

and throughout, the intervention, which more than likely also contributed to the reduction in case counts (Fig 1). A study in northwestern Ethiopia supports this idea. This study reported the highest prevalence of anti-DENV IgM seropositivity during the rainy monsoon and post-monsoon months, suggesting the importance of rainfall on DF transmission and supporting the idea of vector control during these water stagnation periods [21].

## Discussion

Our investigation confirmed a dengue fever outbreak in Kabridahar Town, Somali Region, Ethiopia from May 12–31, 2017. This occurred following prior DF outbreaks in the Somali region in 2014 and the initial introduction of DF into Ethiopia in Dire Dawa City in 2013 [19] (Fig 2B). Since 2013, Ethiopia has reported more than 12,000 dengue fever cases, but this is likely an underestimate due to the lack of Ethiopian IDSR reporting requirements for DF, limited regional and national laboratory capability to confirm cases, and the remoteness of likely endemic areas [31]. This is further underlined by a study conducted in Borea, Ethiopia where acute febrile patients presenting to a hospital were tested serologically for anti-DENV antibodies. Of those tested 25% demonstrated antibodies against DENV infection [32]. This showed that not only is DF a potential threat in the area, but also that the true distribution of the disease in the country may be underestimated. A broader range, country-wide, serological risk assessment for DENV and other flaviviruses demonstrated less dire results, with only 0.5% of samples testing IgG positive for the virus. However, the study found that Ecological Zone 1, the zone into which Kabridahar falls, has a relatively elevated positivity rate for DENV antibodies with results showing 5.3% [33]. Although the outbreak in Kabridahar was unexpected, such information demonstrates a clear explanation, while provides a warning for further potential threats to the area.

Between 1960 and 2010, twenty laboratory-confirmed outbreaks were reported from fifteen African countries, with most occurring in East Africa. All four DENV serotypes (DENV 1,2,3,4) have been isolated in Africa, with DENV 2 reported to cause the most epidemics [13]. Blood sera from a subset of suspected cases from Kabridahar town were tested by RT-PCR for dengue virus, and if positive, for serotype. Positive samples were identified in 72% (15 of 21) of samples tested, all of which were shown to be Dengue serotype 2 (DENV 2). In addition to our study, a recent retrospective health facility-based study performed within Ethiopia's Somali region also revealed DENV 2 circulation [19].

All identified DF cases within this investigation were limited to the urban villages of Kabridahar Town. The highest number of cases, as well as the highest house index (HI), container index (CI), and Breteau index (BI) all occurred in village 07, located in the center of Kabridahar Town. The highest attack rate was also observed in this village. These findings are consistent with other urban outbreaks which displayed similar patterns of disease [7,34]. The neighborhood-matched case-control study identified multiple factors which increased the odds of dengue infection, as well as a protective factor. Due to the strong entomological

component of a dengue infection, the majority of these factors are vector related. Risk factors included a lack of formal education, open containers with *Aedes* larvae, and the wearing of short sleeve t-shirts, while reported use of LLITNs was found protective. A lack of formal education was shown to be significantly associated with the occurrence of DF (AOR = 4.23; CI 1.60–11.17), which is consistent with prior studies [12,31,32]. Previous studies within Ethiopia have also supported this idea. Knowledge of DF preventative measures was absent in approximately 75% of participants in one study [17], while another study showed 87% of participants were unaware of how the virus was transmitted [32]. This lack of education has even encompassed health-care professionals where one study concluded that the "knowledge attitude and practice of health-care professionals were not satisfactory towards dengue fever" [20]. It is quite clear that increasing the country-wide knowledge regarding dengue fever and its prevention could serve beneficial in reducing the future burden of this disease. It has been suggested that more-educated individuals may be more cautious and protective with regard to their immediate environment and health [11,34,35]. It can then be inferred that less-educated persons may be less aware of the importance of disposing or emptying containers that can be used as *Aedes* breeding sites. Education focused on management of standing water near households is warranted in this area.

It was also shown that a majority of DF cases were found in the male population (61%). Similar results were shown in prior studies conducted in Ethiopia and Ecuador [7,19,21,36]. Serological evidence from Borena, Ethiopia supports this as well, demonstrating males had a disproportionally higher rate of DENV IgG when compared to females, 30.1% to 15.4% respectively [32]. The outbreak occurred in a heavily Islamic area of Ethiopia–approximately 98% of the population in the region are Muslim–which may have influenced the gender imbalance in suspected cases. Islamic law requires females to cover their bodies, which may serve as a defensive layer from mosquito bites and an additional form of dengue protection when compared with their male counterparts. This explanation was also suggested as a result of a retrospective health facility-based dengue study in this same Somali Region [19]. Adding further support to this possibility, our study also revealed that those reporting daily wearing of short sleeve T-shirts had a higher likelihood to be a suspected case than those in full dress (AOR = 3.29; CI 1.29–8.39).

Both *Aedes* mosquito species identified in this study are strongly associated with DENV transmission, with *Ae. africanu*s described as a competent vector in maintaining the rural dengue cycle, while *Ae. aegypti* is the main vector in urban areas [37,38]. Although Kabridahar Town is an urban environment, it boarded by a small forested area. The *Ae. africanus* specimens identified were obtained during the entomological larval surveillance of these rural areas. The unplanned urbanization in Kabridahar Town and resulting substandard housing, crowding, and deterioration in water, sewer, and waste management systems have all led to increased breeding habitats and may have contributed to subsequent vector and disease spread. Unplanned urbanization is known to be an important factor for anthropophilic *Aedes* expansion in sub-Saharan Africa [20], with *Ae. aegypti* proving most common in urban environments [39,40]. Open water containers which serve as optimal *Aedes agypti* oviposition sites were seen throughout the outbreak area, resulting in high CI and BI throughout all 9 villages. All of the indices investigated in our study proved significantly higher than those identified in a study conducted in northwest Ethiopia [41] and very similar to those from an entomological investigation that took place in Dire Dawa a year after the 2013 DF outbreak [7]. Both of these studies suggested a resulting high potential for arbovirus transmission. We obtained similar results, as the presence of these containers, both inside and outside the home, was shown to strongly predict dengue infection (Table 1). Uncovered water containers were also shown to be associated with DF and its associated vectors in multiple studies conducted within the

county. The presence of such containers were associated with anti-dengue IgM/IgG antibodies [21], demonstrated to be an independent risk factor for the virus [17], and were shown to be common breeding habitats for the vector [41]. This presence of containers was especially evident in the most effected villages (01–04, 07). The *Ae. aegypti* burden and case counts were highest in the most densely populated section of Kabridahar Town (Village 07).

In response to this, vector control initiatives at oviposition sites, such as the removal of discarded water-holding items, proved effective in reducing the emergence of new adult *Aedes*. However, during these control efforts of the juvenile population, the residual adult *Aedes* population continued to blood feed, before dying off gradually. This is evidenced by the lag in reduction in DF cases after onset of the intervention (Table 1). Additionally, rainfall ceased for more than a month after the onset of the outbreak, which more than likely also contributed to the reduction in oviposition sites and subsequent reduction in emergent adults. Rainfall has long been associated with mosquito activity, and DENV transmission was found to be most active in portions of Ethiopia during monsoon and post-monsoon season [21]. Monthly DENV transmission trends remain limited in Ethiopia as a result of the only recent identification of the infection in country, however in the studies that cite this information, associations with consistent rainfall do suggest it to be of importance [19,21].

In response to this outbreak our team also distributed LLITNs. The use of LLITNs is a known method of reducing human-vector contact from multiple pathogens transmitted by night-biting mosquitoes [42], and was associated with protection from DF in our investigation. Similar findings were also shown regarding DF in Ethiopia and support our findings [17,21]. Female *Ae. aegypti* mosquitoes commonly enter houses at night seeking resting sites. The following morning the mosquito may search out a blood meal from the present household members [43]. Although *Ae. aegypti* are not a night-biting species, the anthropologic characteristics of this community resulted in protection via LLITNs. The response team observed residents of this community commonly participating in early morning and daytime sleeping to avoid the harsh weather associated with this region. As a result, protection from *Ae. aegypti* blood-feeding and possible viral transmission is provided by these nets during the daylight hours. Furthermore, the deployment of LLITN material as window curtains has also been proven to reduce indoor *Ae.aegypti* densities, and theoretically could reduce dengue transmission risk [44,45]. Other protections such as the use of mosquito repellent and indoor residual spraying (IRS) are well-established DF prevention approaches [5,46]. However, we failed to find a similar association in this outbreak. This failure may have been related to the limited access to, or the improper utilization of, repellents, as well as from the lack of IRS application in the prior 6 months.

The case-fatality rate from DF varies by location but can be >20% among those with severe dengue [1,47]. Although severe dengue can be seen in primary infections, this more commonly occurs among persons with prior DF infection of a different serotype (DENV 1,2,3,4) and predominantly among persons under 15 years of age [48]. The low severe dengue rate in this outbreak, along with the corresponding low case-fatality rate (1%), suggests a new introduction of DV into Kabridahar Town, and few, if any prior, infections among the population. Alternatively, it is equally possible that if there were prior dengue infections in the area, they were of this same DENV 2 serotype and therefore would not elicit severe dengue manifestations in the population. Serological testing would help further elucidate these ideas and is under consideration for future efforts. The mechanism of introduction of dengue virus in this area is unknown, but likely related to the travel of infected persons, as 95% of cases and controls reported traveling in the prior 2 weeks. Neighboring Godey Town, approximately 160 km to the southwest of Kabridahar Town, reported DF cases for 3 years prior (2014–2016) to the first infection recorded in Kabridahar (Fig 2B) [19]. It is suspected that the virus, and perhaps even the vector, may have been imported from this area via human movement.

This study did present some limitations. Scarce resources and the lack of a diagnostic laboratory in this rural community limited the ability to laboratory-confirm many of the suspected cases, as well as to ensure that the control population were not asymptomatic carriers. Instead, clinical diagnosis was employed to identify a portion of potential cases and controls. As a result, it is possible that some suspected cases in this study were in fact infected with different, similarly presenting, acute febrile illnesses. This possibility, in conjunction with possible asymptomatic control cases, may slightly impact significant statistical outputs. However, all statistically significant multivariate regression variables demonstrated extremely strong relationships to infection, and it is unlikely this lack of diagnostics would significantly alter the outputs. Additionally, we were only able to collect blood specimens from a portion of DF cases, and of those, some were collected outside the window to confirm DENV by RT-PCR. Further resource limitations at the laboratory level prevented the sequencing of identified positive samples. As of the publishing of this manuscript, there are no sequencing facilities or capacity within Ethiopia. Identifying the sequence of the circulating dengue strain would have proven valuable for tracking the movement and evolution of the virus, particularly if compared to samples from the prior outbreaks. All positive samples are stored at -80˚C with the plan to undergo sequencing when such capacities are in place.

During this investigation, additional limitations presented themselves entomologically, as well as with data availability. It is possible that the sampled adult *Aedes* may not have represented the true composition of Kabridahar's *Aedes* population during the time of investigation, as only a subset of households within this large community were entomologically investigated. We were also unable to isolate DENV from any of the adult *Aedes* mosquito specimens. Finally, we were unable to acquire a map of population denominators for the villages in the district, which prevented a more nuanced look at disease spread and attack rates.

## Conclusion

Our epidemiological, entomological, and laboratory investigation confirmed a DF outbreak in Kabridahar Town. Higher odds of being a suspected DF case were seen among those without formal education, who wore short-sleeved shirts, and had exposure to open containers with larvae; LLITN provided DF protection. In the long term, we recommend improved vector surveillance and control programs, promoting best practices in preserving water, and disposal of containers in reducing *Aedes* density. Similar recommendations have been made as a result of previous DF investigations within Ethiopia [17,19,21], as well as in a continent-wide systematic review of DENV infection [15], and we echo these in an effort to prevent and manage future outbreaks. Additionally, healthcare worker education is recommended, which is also supported by findings from this same continent-wide systematic review [15]. This approach may lead to earlier identification of outbreaks, better case management, additional sample collection, and strengthened surveillance.

## Supporting information

**S1 Table. Entomological Survey—Kabridahar District, Korahay Zone, Somali Region, Ethiopia, 2017.**
(DOCX)

## Acknowledgments

The authors wish to thank Ethiopian Public Health Institute, Public Health Emergency Management Directorate for logistical and technical support to carry out this investigation. They

also would like to thank the Somali regional Health Bureau, Korahey Zonal Health Department, and Kabridahar District Health Offices for their facilitation during the investigation.

## Author Contributions

**Conceptualization:** Mulugeta Asefa Gutu.

**Data curation:** Mulugeta Asefa Gutu.

**Formal analysis:** Mulugeta Asefa Gutu.

**Investigation:** Mulugeta Asefa Gutu, Yusuf Mohammed, Fekadu Gemechu, Adamu Tayachew, Lehageru Gizachew, Moti Idosa.

**Methodology:** Mulugeta Asefa Gutu.

**Project administration:** Mulugeta Asefa Gutu, Abyot Bekele Woyessa, Yohanis Dugasa.

**Resources:** Mulugeta Asefa Gutu.

**Software:** Mulugeta Asefa Gutu.

**Supervision:** Mulugeta Asefa Gutu, Abyot Bekele Woyessa, Ryan E. Tokarz, David Sugerman.

**Validation:** Mulugeta Asefa Gutu.

**Visualization:** Mulugeta Asefa Gutu.

**Writing – original draft:** Mulugeta Asefa Gutu.

**Writing – review & editing:** Mulugeta Asefa Gutu, Alemayehu Bekele, Yimer Seid, Ryan E. Tokarz, David Sugerman.

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
