## [Decision Letter · Decision Letter 0]

2 Jun 2020

Dear Mr. Gutu,

Thank you very much for submitting your manuscript "Another Dengue Fever Outbreak in Eastern Ethiopia – An Emerging Public Health Threat" for consideration at PLOS Neglected Tropical Diseases. As with all papers reviewed by the journal, your manuscript was reviewed by members of the editorial board and by several independent reviewers. In light of the reviews (below this email), we would like to invite the resubmission of a significantly-revised version that takes into account the reviewers' comments. 

Dear Authors,

Our apologies for this long process. It has been harder than usual to find reviewers and to get reviews in time.

I had to make the hard decision to unassign our third reviewer, as he was more than 15 days late.

We hope you will find the reviewers' comments useful.

Kind regards,

Elvina Viennet

We cannot make any decision about publication until we have seen the revised manuscript and your response to the reviewers' comments. Your revised manuscript is also likely to be sent to reviewers for further evaluation.

Sincerely,

Elvina Viennet, PhD

Deputy Editor

Francis Jiggins

Deputy Editor

Dear Authors,

Our apologies for this long process. It has been harder than usual to find reviewers and to get reviews in time.

I had to make the hard decision to unassign our third reviewer, as he was more than 15 days late.

We hope you will find the reviewers' comments useful.

Kind regards,

Elvina Viennet

Reviewer's Responses to Questions

**Key Review Criteria Required for Acceptance?**

**Methods**

-Are the objectives of the study clearly articulated with a clear testable hypothesis stated?

-Is the study design appropriate to address the stated objectives?

-Is the population clearly described and appropriate for the hypothesis being tested?

-Is the sample size sufficient to ensure adequate power to address the hypothesis being tested?

-Were correct statistical analysis used to support conclusions?

-Are there concerns about ethical or regulatory requirements being met?

Reviewer #1: The author described, the RT-PCR Assay provided by CDC. Stating this is not relevant.

Reviewer #2: The objectives of the study are clearly articulated and the study design is appropriate.

Correct statistical analysis were used to support conclusions

However, details of the laboratory methods are required to better undestand the outbreak.

**Results**

-Does the analysis presented match the analysis plan?

-Are the results clearly and completely presented?

-Are the figures (Tables, Images) of sufficient quality for clarity?

Reviewer #1: Describe the relationship between rainfall and outbreak i.e. after how many weeks of rainfall end up/reduced did the outbreak started and the like.

The first sentence of the last paragraph said "In response to the epidemic, from May 15 – June 17, 2007". Is this year correct?

Reviewer #2: The analysis presented matches the analysis plan and the results are clearly but not completely presented as details on the index case is missing as well as details on the severe cases and/or death case together with their laboratory results.

Figure 2 should highlight the location of previous outbreaks.

**Conclusions**

-Are the conclusions supported by the data presented?

-Are the limitations of analysis clearly described?

-Do the authors discuss how these data can be helpful to advance our understanding of the topic under study?

-Is public health relevance addressed?

Reviewer #1: The recommendation of health care worker education is not supported by the findings in this study.

Reviewer #2: The conclusion are supported by the data presented and the limitations are clearly described.

The public health relevance of the study has been addressed. However, discussion of these data compare to other available and published data in the country is missing.

**Editorial and Data Presentation Modifications?**

Reviewer #1: (No Response)

Reviewer #2: A few major revisions are needed.

**Summary and General Comments**

Reviewer #1: (No Response)

Reviewer #2: The study was overall well conducted although several major points need to be addressed. Authors are strongly advice to do sequencing of the circulating dengue strain as it is important for tracking the movement and evolution of these viruses.

PLOS authors have the option to publish the peer review history of their article (what does this mean?). If published, this will include your full peer review and any attached files.

Reviewer #1: No

Reviewer #2: Yes: Maurice Demanou
---

## [Decision Letter · Decision Letter 1]

11 Oct 2020

Dear Mr. Gutu,

Thank you very much for submitting your manuscript "Another Dengue Fever Outbreak in Eastern Ethiopia – An Emerging Public Health Threat" for consideration at PLOS Neglected Tropical Diseases. As with all papers reviewed by the journal, your manuscript was reviewed by members of the editorial board and by several independent reviewers. The reviewers appreciated the attention to an important topic. Based on the reviews, we are likely to accept this manuscript for publication, providing that you modify the manuscript according to the review recommendations. 

Sincerely,

Elvina Viennet, PhD

Deputy Editor

Francis Jiggins

Deputy Editor

Reviewer's Responses to Questions

**Key Review Criteria Required for Acceptance?**

**Methods**

-Are the objectives of the study clearly articulated with a clear testable hypothesis stated?

-Is the study design appropriate to address the stated objectives?

-Is the population clearly described and appropriate for the hypothesis being tested?

-Is the sample size sufficient to ensure adequate power to address the hypothesis being tested?

-Were correct statistical analysis used to support conclusions?

-Are there concerns about ethical or regulatory requirements being met?

Reviewer #1: The objective of the study is clearly stated. The study design is appropriate and the study population is described well. The sample size is sufficient to infer the findings. Correct statistical analysis computed.

Reviewer #2: (No Response)

**Results**

-Does the analysis presented match the analysis plan?

-Are the results clearly and completely presented?

-Are the figures (Tables, Images) of sufficient quality for clarity?

Reviewer #1: The findings are presented clearly with good quality and clear tables and figures.

Reviewer #2: (No Response)

**Conclusions**

-Are the conclusions supported by the data presented?

-Are the limitations of analysis clearly described?

-Do the authors discuss how these data can be helpful to advance our understanding of the topic under study?

-Is public health relevance addressed?

Reviewer #1: The conclusions are based on the findings, limitations of the study and public health relevance stated well.

Reviewer #2: (No Response)

**Editorial and Data Presentation Modifications?**

Reviewer #1: (No Response)

Reviewer #2: (No Response)

**Summary and General Comments**

Reviewer #1: (No Response)

Reviewer #2: (No Response)

PLOS authors have the option to publish the peer review history of their article (what does this mean?). If published, this will include your full peer review and any attached files.

Reviewer #1: Yes: Mikias Alayu Alemu

Reviewer #2: Yes: Maurice Demanou
---

## [Editor Report · Decision Letter 2]

17 Nov 2020

Dear Mr. Gutu,

We are pleased to inform you that your manuscript 'Another Dengue Fever Outbreak in Eastern Ethiopia – An Emerging Public Health Threat' has been provisionally accepted for publication in PLOS Neglected Tropical Diseases.

Best regards,

Elvina Viennet, PhD

Deputy Editor

Francis Jiggins

Deputy Editor

---

## [Editor Report · Acceptance letter]

12 Jan 2021

Dear Mr. Gutu,

We are delighted to inform you that your manuscript, "Another Dengue Fever Outbreak in Eastern Ethiopia – An Emerging Public Health Threat," has been formally accepted for publication in PLOS Neglected Tropical Diseases.

Best regards,

Shaden Kamhawi

co-Editor-in-Chief

Paul Brindley

co-Editor-in-Chief
